

**FZStats v1.0: a raster statistics toolbox for simultaneous management of spatial**
**stratified heterogeneity and positional dependence in Python**
Na Ren[1], Daojun Zhang[2*], and Qiuming Cheng[1, 3]
[1]School of Earth Resources, China University of Geosciences, Wuhan, 430043, China
[2]School of Public Administration, China University of Geosciences, Wuhan 430074, China
[3]State Key Laboratory of Geological Processes and Mineral Resources, China University of
Geosciences, Wuhan 430043, China
*Corresponding author: cugzdj@gmail.com (Zhang, D)
**Abstract:** Based on the traditional Focal Statistics and Zonal Statistics tools of mainstream
GIS software, we developed a raster statistics toolbox named FZStats v1.0 using Python3 and
QT5. The main contributions of this study are as follows. Firstly, the development of a
specialized spatial analysis toolset designed to comprehensively address stratified
heterogeneity, positional dependence, and their combinations, thereby addressing gaps in
existing Focal and Zonal methods that individually tackle stratified heterogeneity and
positional dependence problems. Secondly, our toolset features a user-friendly interface and
structure, integrates both existing and enhanced spatial statistical methods, supports
multi-processing and batch processing capabilities, and provides users with the flexibility to
select calculation methods tailored to their computer configurations and application
requirements. Thirdly, the newly proposed Focal-Zonal Mixed Statistics method demonstrates
superior predictive accuracy compared to the traditional Focal Statistics and Zonal Statistics
methods in geothermal detection, which preliminarily showcases the advantages of this new
approach. Additionally, we discussed the advantages, robustness, and advancements of the
Focal-Zonal Mixed Statistics method, concluding that the development of this new method
and toolset is necessary and holds substantial potential for applications across diverse fields.
**Keywords:** Spatial Statistics; Raster Operations; Spatial Heterogeneity; Spatial Dependency;
Focal/Zonal Statistics.





## 1 Introduction

The advent of Geographic Information Systems (GIS) marks a milestone in the evolution of geography. As a core function of GIS software, spatial statistics provide powerful methods and tools that enable researchers and decision-makers to analyze spatial patterns and associations on the Earth's surface comprehensively and accurately. Spatial heterogeneity and positional dependence are two fundamental characteristics to be considered in spatial data processing (Goodchild and Haining, 2004). Correspondingly, Zonal Statistics and Focal Statistics are two essential methods of spatial statistical analysis. The former can be achieved through a model that involves partitioning raster data into several zones based on predefined rules or attributes, performing statistical analyses on the raster cells within each zone, and then outputting the results as a mosaic raster layer (Singla and Eldawy, 2018; Haag et al., 2020; Winsemius and Braaten, 2024). The latter, also known as neighborhood or local window statistics, takes each raster cell as the center and extends a specified range surrounding the center to form a local window according to the designated window size; it performs statistical analyses on the raster cells within this window and then outputs the results as a mosaic raster layer (Mathews and Jensen, 2012; Kassawmar et al., 2019; Zhang et al., 2021). The calculated statistics for both zonal and focal methods are similar, including the mean, maximum, minimum, sum, and so on.

Currently, the mainstream GIS software platforms including ArcGIS and QGIS provide tool modules such as Focal Statistics and Zonal Statistics, which have promoted the usage of these two methods. From an application perspective, Zonal Statistics primarily address spatial stratified heterogeneity (SSH), which can be detected by dividing the target variable though environmental characteristic classified variables (Wang et al., 2016; Wang and Xu, 2017; Gao et al., 2022). For instance, the actual or potential growth of vegetation may vary significantly due to different environmental conditions such as slope and aspect (Zhang et al., 2018, 2019;



Xu et al., 2020). With respect to Focal Statistics, it focuses on spatial position dependence
(SPD), which can be addressed or at least weaken by introducing the local windows or
geographic weights (Tobler, 1970; Wolter et al., 2009; Wagner et al., 2018). For example,
even soils or rocks with the same texture generally exhibit variations in geochemical element
content due to their different spatial locations; however, these differences diminish with
decreasing distance, indicating that these attributes are dependent on spatial position (Krige
and Magri, 1982; Trangmar et al., 1986; Zuo, 2014).

In our real world, SSH and SPD may coexist, with the former exhibiting abrupt changes

and the latter exhibiting gradual changes. For example, due to variations in land-sea
distribution, solar radiation, and altitude, terrestrial vegetation exhibits strong meridional,
latitudinal, and vertical zonal distribution patterns respectively (Qiu et al., 2013; Dong et al.,
2019; Eddin and Gall, 2024), which explains the significant SPD in vegetation coverage.
Meanwhile, due to the influence of local topography, microclimate, and human activities, the
vegetation coverage differences caused by these factors do not entirely manifest as gradual
changes. Typical evidence includes phenomena such as vegetation on shady and sunny slopes
generally shows SSH (Álvarez-Martínez et al., 2014; Zhang and Zhang, 2022; ) and
significant differences between urban and rural landscapes (Zhang et al., 2023b). Furthermore,
due to differences in formation age, there are significant variations in material across strata,
which is a major reason for the SSH of mineral resources distribution (Zhao Pengda, 2006;
Zuo, 2020). Subsequently, under the influence of internal and external geological processes,
the distribution of mineralization elements often exhibits SPD characteristics (Cheng, 2006,
2012), and Geostatistics and Kriging methods were developed to explain this phenomenon
(Krige, 1951; Goovaerts, 1997; Müller et al., 2022). Therefore, when dealing with problems
involving spatial statistics, it is necessary to consider both SSH and SPD simultaneously.

Some scholars have noted this issue and developed certain improved models in their



respective fields to overcome the challenges posed by solely considering SSH or SPD.
Professor Zhu and his group expanded upon traditional spatial interpolation methods, which
typically focus solely on spatial dependence, by introducing constraints derived from
environmental similarity (Zhu et al., 2019). They further proposed the "Third Law of
Geography", which states that the more similar the geographic configurations of locations, the
more similar the values (processes) of the target variable at these locations (Zhu et al., 2018;
Zhu et al., 2020). Meanwhile, Professor Zhang and his group enhanced traditional vegetation
potential assessment models, which typically only consider similar habitat conditions, by
incorporating spatial sliding window techniques (Zhang et al., 2019). This development led to
a model for assessing vegetation restoration potential based on local windows, simultaneously
considering spatial proximity and environmental similarity (Xu et al., 2020; Zhang, 2023a).
The most recent attempt at spatial statistical modeling that considers both SSH and SPD is by
Lessani and Li (2024), who developed similarity and geographically weighted regression
model. This new model integrates distance weights and similarity weights to address the
limitations of traditional geographically weighted modeling that only considers spatial
dependency.
These studies focused on specific issues such as spatial interpolation, regression, and
extreme values. Although these models effectively address the combination of both SSH and
SPD, there is currently a lack of a universal spatial statistics tool similar to Focal Statistics
and Zonal Statistics. This study aims to develop a spatial statistical model, termed the
Focal-Zonal Mixed Statistics, within the framework of GIS spatial statistics. The newly
developed toolbox, FZStats v1.0, integrates traditional Focal Statistics and Zonal Statistics, as
well as Focal-Zonal Mixed Statistics. In terms of algorithm design, we employ
multiprocessing and batch processing techniques, which promise to enhance operational
efficiency and user experience. We believe that the FZStats v1.0 toolbox, especially the newly



proposed Focal-Zonal Mixed Statistics, has the potential to offer methods and tools to better
understand and address SSH and SPD issues.
**2 Models**
**2.1 Focal Statistics model**
The modeling of Focal Statistics involves three functional methods: (1) defining the
neighborhood window, (2) identifying the cells located within the neighborhood, and (3)
calculating the neighborhood statistics.
**2.1.1 Defining the neighborhood window**
Defining the neighborhood window is a crucial prerequisite for Focal Statistics. There are two
parameters to define the neighborhood window: its shape and size. These can be adjusted
based on the spatial characteristics of the data and the objectives of the research. Commonly
used shapes include circular, square, and rectangular, while the size is typically specified in
terms of the number of cells.
Formally, let $NW$ denote the neighborhood window, the following expression can be
obtained.
$NW = f(Shape,\ Size)$            (1)
where $f(.)$ represents the function used to characterize the neighborhood window, $Shape$
refers to the geometric configuration of the window, while $Size$ specifies its extent.
**2.1.2 Identifying cells within the neighborhood**
Once the neighborhood window is determined, the spatial sliding window technique can be
used to identify the cells located within the neighborhoods defined by the neighborhood
window centered around given cells (Hyndman and Fan, 1996). For each current location
$Cell(i, j)$, the neighborhood can be expressed as:
$Nbh(i, j) = nbh(Cell(i, j),\ NW)$            (2)
where $i$ and $j$ denote the row and column number of current cell at location $(i, j)$, respectively;





$nbh(.)$ is the function for determining the neighborhood of $Cell(i, j)$, and $NW$ represents the
neighborhood window.
Then cells located within $Nbh(i, j)$ form a cell set, which can be described as follows:
$CS_F(i, j) = \{Cell(i', j') \in \mathbf{R}_v \mid is\_in\_nbh(Cell(i', j'), Nbh(i, j)) == TRUE\}$   (3)
where $is\_in\_nbh(.)$ is the indicator function used to identify whether $Cell(i', j')$ is located
within the neighborhood $Nbh(i, j)$; $i'$ and $j'$ are for the row and column number of the
input value raster $\mathbf{R}_v$, respectively.
In Eq. (3), the detailed form of $is\_in\_nbh(.)$ depends on the shape of the neighborhood
window. For example, when the window is circular, $is\_in\_nbh(.)$ can be expressed as:
$\sqrt{(i' - i)^2 + (j' - j)^2} \leq d$   (4)
where $d$ is the radius of the circular window, i.e. window size, and $i$ and $j$, and $i'$ and $j'$ are
as explained above.
**2.1.3 Calculating the focal statistics**
Suppose that $ST_F(Type, Set)$ denotes the statistical function of Focal Statistics, and $Type$
and $Set$ are for the statistical parameter and the cell set to be processed. At the location of
$Cell(i, j)$ and under the Focal Statistics model, $Set$ can be specified as $CS_F(i, j)$. Then the
output of the Focal Statistics for $Cell(i, j)$ can be expressed as:
$O_F(i, j) = ST_F(Type, CS_F(i, j))$   (5)
Expressed in terms of raster layer operations, Eq. (5) can be further formulated as:
$\mathbf{R}_{F\_out} = Focal\_Statistics(\mathbf{R}_v, NW, Type)$   (6)
where $\mathbf{R}_v$ and $\mathbf{R}_{F\_out}$ represent the input value raster and the output raster for Focal
Statistics, respectively, while $NW$ and $Type$ denote the functions for neighborhood window
and statistical type in that order.
**2.2 Zonal Statistics model**
Unlike Focal Statistics, which require only a value raster as input, Zonal Statistics require two



input raster layers: one as the value raster and the other as the zone raster. The zone raster
defines the shape and distribution of the zones, and each cell can only belong to a single zone.
Zonal Statistics calculates the statistics for each zone based on the corresponding cells from
the value raster, and the calculated statistic is assigned as the output value for all cells within
the zone. Finally, the output values of different zones are assembled into the output raster.

Zonal Statistics modeling involves two functional methods, which are for identifying the

cells in the value raster by zone and calculating zonal statistics respectively.
**2.2.1 Identifying cells in the value raster falling into each zone**
In Zonal Statistics, spatial overlay analysis can be used to find the zone code for each cell in
the value raster (Hyndman and Fan, 1996):
$Z_k(i', j') = Zone(Cell(i', j'))$        (7)
where $Z_k(i', j')$ represents the zone code at location $(i', j')$, and $Zone(.)$ is the function that
returns the zone code for the value raster cell at location $(i', j')$.

For a given zone $Z_k$, the corresponding cells in the value raster form a cell set $CS_Z(Z_k)$,

which can be expressed as:
$CS_Z(Z_k) = \{ Cell(i', j') \in R_v \mid Zone(Cell(i', j')) == Z_k \}$        (8)
**2.2.2 Calculating the zonal statistics**
The calculation of statistics for a given zone $Z_k$ can be represented as:
$O_Z(Z_k) = ST_Z(Type, CS_Z(Z_k))$        (9)

It is important to note that the calculated statistics are assigned to all cells within each

zone, and the statistics for all zones are ultimately mosaicked into the output raster.

Using $R_v$ and $R_Z$ to denote the input layers of value raster and zone raster,

respectively. Zonal Statistics can be expressed as:
$R_{Z\_out} = Zonal\_Statistics(R_v, R_z, Type)$        (10)
where $R_{Z\_out}$ represents the output raster, and $Type$ is for the statistic type.





**2.3 Focal-Zonal Mixed Statistics**
Similar to Zonal Statistics, Focal-Zonal Mixed Statistics also require two input raster layers,
and the specific modeling process involves the following two functional methods.
**2.3.1 Identifying cells within the neighborhood that belong to the same zone**
Actually the determination of the target cell set in Focal-Zonal Mixed Statistics combines
both the spatial proximity condition from Focal Statistics, and the environmental
characteristic similarity condition from Zonal Statistics. For $Cell(i, j)$ at the current location,
if its neighborhood is $Nbh(i, j)$ and its zone code is $Z_k(i, j)$, then its cell set consists of all
cells within the neighborhood that belong to the same zone as the cell in Focal-Zonal Mixed
Statistics. Mathematically, this can be expressed as:
$$CS_{F-Z}(i,j) = \left\{ Cell(i', j') \in \boldsymbol{R}_v \left| \begin{array}{c} is\_in\_nbh\left(Cell(i', j'),\ Nbh(i, j)\right) == TRUE \\ Zone\left(Cell(i', j')\right) == Z_k(i, j) \end{array} \right. \right\} \qquad (11)$$
**2.3.2 Calculating the focal-zonal mixed statistics**
Still using $Type$ to represent the statistical type, the output result of Focal-Zonal Mixed
Statistics for the current $Cell(i, j)$ can be expressed as:
$$O_{F-Z}(i,j) = ST_{F-Z}(Type, CS_{F-Z}(i,j)) \qquad (12)$$

In the form of raster layer operations, Eq. (12) can be further expressed as:

$$\boldsymbol{R}_{FZ\_out} = Focal\_Zonal\_Statistics(\boldsymbol{R}_v,\ \boldsymbol{R}_z,\ NW,\ Type) \qquad (13)$$
where $\boldsymbol{R}_v$, $\boldsymbol{R}_z$, and $\boldsymbol{R}_{FZ\_out}$ represent the value raster, zone raster, and output raster for
Focal-Zonal Mixed Statistics, respectively; $NW$ is the neighborhood window, and $Type$ is
for statistical parameter.
**3 Module design**
**3.1 Modeling process for Focal-Zonal Mixed Statistics**
The flowchart for the newly proposed Focal-Zonal Mixed Statistics is presented in Fig. 1, and
the detailed modeling process is described as follows.



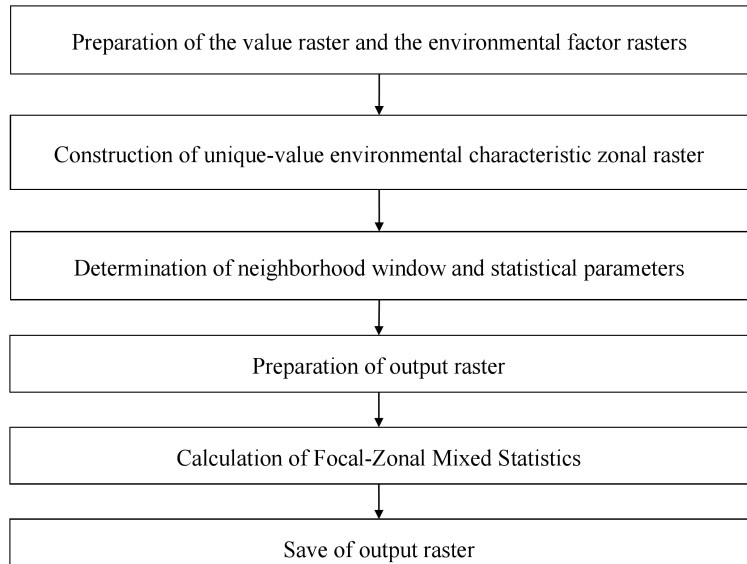


**Figure 1**. Flowchart for the modeling of Focal-Zonal Mixed Statistics


(1) Preparation of the value raster and the environmental factor rasters
This initial step involves collecting and preprocessing the spatial data required for the
analysis. The value raster typically represents the primary variable of interest, i.e., the target
layer, such as temperature, pollution levels, or vegetation indices. Environmental factor
rasters include various influencing factors, such as elevation, slope, land cover, and other
relevant geographical features that may contribute to the heterogeneous distribution of the
target layer. Preprocessing methods may include resampling, reprojecting, and normalizing
the data to ensure consistency and compatibility among the raster layers, so that they share the
same spatial extent, resolution, and reference system.
(2) Construction of unique-value environmental characteristic zonal raster (UV-ECZR)
This process can be achieved using the "Reclassify" tool in ArcGIS to transform
continuous or categorical environmental factor rasters into discrete classes based on
predefined criteria. Subsequently, the UV-ECZR is generated through spatial overlay analysis
and unique-value encoding. Cells in the UV-ECZR that share the same unique-value



environmental characteristic code (UV-ECC) form a similar environmental unit (SEU). A
detailed implementation of this process is described in the following Sect. 3.2.1.
(3) Determination of neighborhood window and statistical parameters
This process involves defining the neighborhood window and specifying the statistical
parameters for Focal-Zonal Mixed Statistics.
(4) Preparation of output raster
This step involves creating an output raster with the same spatial extent, resolution, and
reference system as the input rasters. This output raster will store the results of the
Focal-Zonal Mixed Statistics calculations.
(5) Calculation of the statistics
In this step, the moving window technique is applied to locate each current cell and its
local window. For each current cell, identify the neighborhood cells based on the defined
neighborhood window parameters (refer to Sect. 2.1.1). Within this neighborhood, isolate the
cells within the same SEU as the current cell. Subsequently, calculate the specified statistic
for these cells in the value raster that correspond to those isolated cells.
(6) Save of output raster
Write the statistical result to each corresponding cell in the output raster one at a time,
and save the raster file after all cells have been processed.
The core algorithm involved in the above steps is described in the following section.
**3.2 Core algorithm design for Focal-Zonal Mixed Statistics**
**3.2.1 Algorithm design for the UV-ECZR construction**
Assume that there are $p$ continuous environmental variables, i.e., $E_1, E_2, \ldots, E_p$, with their
corresponding reclassified variables being $CE_1, CE_2, \ldots, CE_p$. The number of categories and
the digit lengths of these categories are denoted as $S_1, S_2, \ldots, S_p$ and $D_1, D_2, \ldots, D_p$,
respectively. The method for calculating the digit lengths of the categories is as follows:



$$D_q = \lfloor \lg S_q \rfloor + 1 \tag{14}$$
where lg denotes the logarithm with base 10, $\lfloor . \rfloor$ represents the floor function, and $q =$
$1, 2, \ldots, p$. The categories for the $q$-th environmental variable should be a positive integer,
and the range of cell value in the reclassified raster ($\boldsymbol{CE}_q$) can be expressed as $[1, S_q]$.
Then, the UV-ECC at location $(i, j)$ can be defined as:
$$UV - ECC\,(i, j) = 1\overbrace{X\cdots X}^{D_1}\overbrace{X\cdots X}^{D_2}\cdots\overbrace{X\cdots X}^{D_q}\cdots\overbrace{X\cdots X}^{D_p} \tag{15}$$
where $\overbrace{X\cdots X}^{D_q}$ represents the category code of $\boldsymbol{CE}_q$ at location $(i, j)$, $D_q$ is obtained through
Eq. (14). To keep the consistency in the UV-ECC format, it is necessary to prepend a
sufficient number of "0"s to ensure the digit length of category code equals $D_q$.
In the form of raster calculator, the UV-ECZR can be expressed as:
$$UV - ECZR = CE_1 \cup CE_2 \cup \ldots \cup CE_p \tag{16}$$
where $\cup$ represents the spatial overlay.
**3.2.2 Algorithm design for determining the valid range for statistics under the sliding**
**window technique**
A rectangular window, which aligns with the rows and columns of raster data and is both easy
and efficient to implement, is commonly used in the sliding window technique. However, its
drawback is also evident: the grid cells at the four corners are much farther from the current
location than those on the horizontal and vertical axes (Zhang et al., 2016a). Despite this,
rectangular windows remain one of the most popular forms of spatial sliding windows. In this
study, we consider rectangular windows along with circular and elliptical windows. Since a
circle is a special form of an ellipse, we use the ellipse as an example to illustrate the
algorithm design for determining the valid range for statistics under the sliding window
technique in Focal-Zonal Mixed Statistics.
(1) Mask matrix for elliptical window



An elliptical window is defined by three key parameters: the length of major axis, the
ratio of the minor axis to the major axis, and the deflection angle of major axis. Let $(x_0, y_0)$
represent the center of the ellipse, i.e., the current location, $a$ denotes the semi-major axis
length, $r$ be the minor-to-major axis ratio, and $\theta$ be the deflection angle. Then the elliptical
window can be mathematically expressed as:
$Ellipse((x_0, y_0), a, r, \theta) = \dfrac{[(x-x_0)\cos\theta + (y-y_0)\sin\theta]^2}{a^2} + \dfrac{[-(x-x_0)\sin\theta + (y-y_0)\cos\theta]^2}{(ra)^2}$     (17)
Based on Eq. (15), the bounding box of the elliptical window can be represented as
$BBox_{ellipse}(minX, maxX, minY, maxY)$, where $minX, maxX, minY, maxY$ are as follows:
$\begin{cases} minX, maxX = x_0 \pm \sqrt{\dfrac{4CF}{B^2 - 4AC}} \\ minY, maxY = y_0 \pm \sqrt{\dfrac{4AF}{B^2 - 4AC}} \end{cases}$     (18)
here,
$\begin{cases} A = a^2(\sin^2\theta + r^2\cos^2\theta) \\ B = 2a^2(r^2 - 1)\sin\theta\cos\theta \\ C = a^2(\cos^2\theta + r^2\sin^2\theta) \\ F = -\frac{1}{2}(Dx_0 + Ey_0) - r^2a^4 \end{cases}$     (19)
The bounding box $BBox_{ellipse}$ provides a simplified and direct spatial reference for
constructing a Boolean mask matrix for the elliptical window, i.e., $Matrix_{Ellipse\_mask}$, where
cells inside and outside the $BBox_{ellipse}$ are assigned values of "True" and "False",
respectively. In Focal Statistics, this mask is used directly to define the area of interest for
statistics, see Fig. 2a.

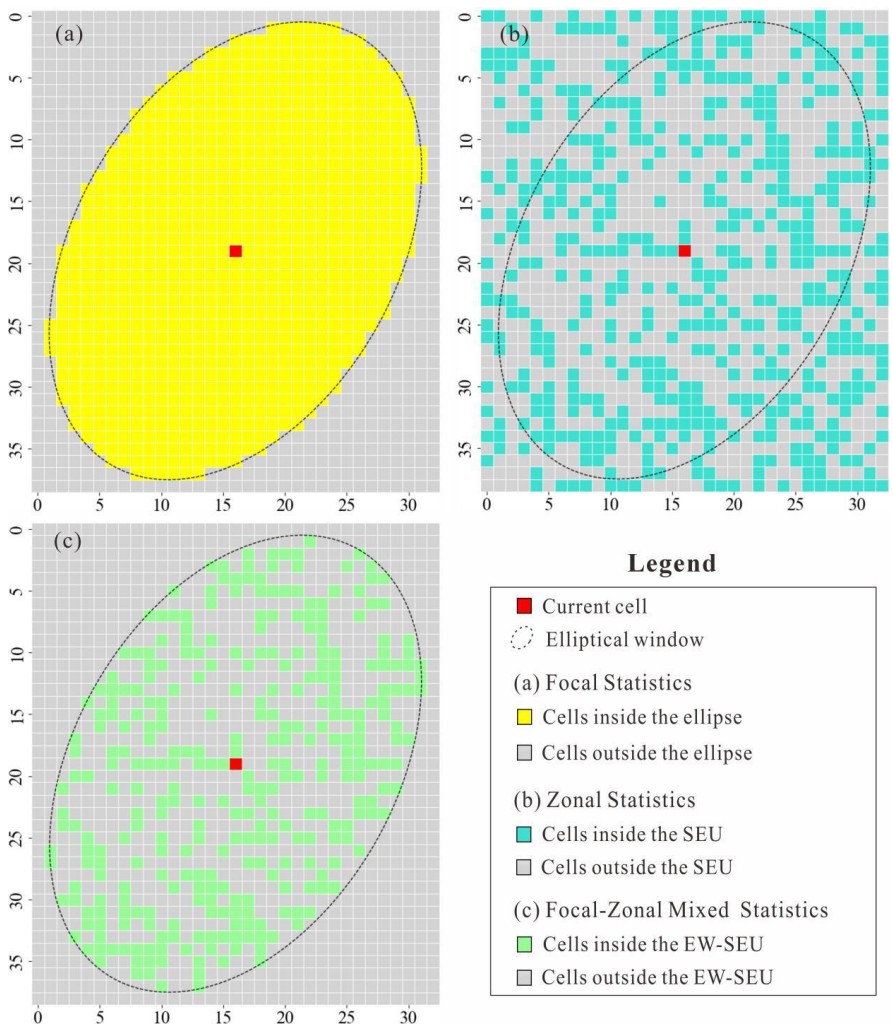

**Figure 2.** Heatmaps for the Boolean mask matrix: (a) the elliptical window of Focal Statistics, (b) the

similar environmental unit (SEU) of Zonal Statistics, and (c) the elliptical window similar environmental

unit (EW-SEU) of Focal-Zonal Mixed Statistics

(2) Mask matrix for similar environment in the bounding box

SEU is the basic object of Zonal Statistics. In Focal-Zonal Mixed Statistics, for the

current cell, the elliptical window similar environmental unit (EW-SEU) is established

according to the environmental characteristic code within the initial neighborhood window

defined by the bounding box. Using $Matrix_{Similarity\_mask}$ to represent this unit, cells with



the same environmental characteristic code as the current cell are assigned a value of "True",
while others are assigned a value of "False", as shown in Fig. 2b.
(3) Mask matrix for similar environment in the elliptical window
The matrices of steps (1) and (2) shares the same dimensions, and thus the similar
environment mask matrix for the current cell in the elliptical window can be constructed using
a logical "AND" operation between these two matrices, as expressed in the following
equation:
$Matrix_{E\_S\_mask} = Matrix_{Similarity\_mask} \wedge Matrix_{Ellipse\_mask}$ (20)
where $\wedge$ denotes the logical "AND" operator. $Matrix_{E\_S\_mask}$ serves as the basis for
determining the valid range for Focal-Zonal Mixed Statistics, as illustrated in Fig. 2c.
**3.2.3 Algorithm design for the statistics calculation**
The algorithm for the statistics calculation is designed as follows:
(1) Determination of valid statistical cells in the value raster
Using $Matrix_{Value}$ to represent the cell array from the value raster within the bounding
box defined above, then by performing a bitwise multiplication of $Matrix_{E\_S\_mask}$ with
$Matrix_{Value}$, the final valid statistical value matrix $Matrix_{Valid}$ is obtained:
$Matrix_{Valid} = Matrix_{E\_S\_mask} \otimes Matrix_{Value}$ (21)
where $\otimes$ denotes bitwise multiplication. This operation collects cells from the value raster that
are located within the neighborhood and share the same UV-ECC as the current cell, while
masking out other cells that could interfere with the statistical results. In $Matrix_{Valid}$, the
masked cells can be represented with "NaN".
(2) Design of the calculation function for the statistics
Taking $Matrix_{Valid}$ as the final input, the calculation functions for Focal-Zonal Mixed
Statistics can be designed based on scientific computing tools such as NumPy. This library
provides a range of statistical methods, including minimum, maximum, mean, standard



deviation, percentiles, and more. For instance, the "*numpy.nanmax*()" method can ignore
"NaN" values and return the maximum value of $Matrix_{Valid}$, while the
"*numpy.nanpercentile*()" method, also ignoring "NaN" values, calculates the n-th percentile
of $Matrix_{Valid}$.
**3.3 User interface design**
The Focal-Zonal Mixed Statistics, along with traditional Zonal Statistics and Focal Statistics,
are included in the newly developed toolbox, FZStats v1.0, using Python3 and QT5. The user
interface is organized into three tabs, each dedicated to one of the three methods, allowing
users to switch among them (see Fig. 3). Taking the tab for Focal-Zonal Mixed Statistics as an
example, the interface is divided into four main sections, and the detailed description of the
user interface design is given as follows.

(1) Input and output design

Users can select the value raster and UV-ECZR as input data from their datasets.

Additionally, they can specify the output path and filename for the resulting raster data.

(2) Neighborhood window design

Users can configure the shape (e.g., rectangular, circular, elliptical) and size (e.g.,

number of cells or spatial units) of the neighborhood window. For rectangular and circular
windows, size is specified by the half-side length and radius, respectively. Elliptical windows
are characterized using three morphological parameters: the length of the major axis, the ratio
of the minor axis to the major axis, and the deflection angle of major axis.

(3) Statistical measure design

Users can select a specified statistical measure from the dropdown menu. For percentile

calculations, users are required to specify the exact percentile values of interest, such as the
50th, 75th, or 98th percentiles.

(4) Optimization settings



In this section, users can fine-tune various parameters to optimize the calculation
performance. Key settings include:
Chunk processing: Users can divide the input raster into smaller chunks, which can
enhance performance by reducing the memory load and making it easier to handle large
datasets.
Parallel processing: Users can configure the number of processors used for parallel
processing to reduce computation time. On computers with higher configurations, increasing
the number of processors allows for the utilization of more cores, enabling simultaneous task
execution and significantly reducing processing times.
Threshold setting: Users can specify a minimum sample threshold for statistical
calculations, which defines the minimum number of cells required for performing the
statistical measure. This threshold ensures that the statistical computations are based on a
sufficient sample size, thereby enhancing the reliability and robustness of the results.
Additionally, to further improve multitasking efficiency and achieve a certain degree of
automation, a batch processing feature is provided in the toolbox. Users can define parameters
in an INI-format configuration file (config.ini), which simplifies the process by eliminating
repetitive configurations. This feature allows users to set up and execute multiple tasks in a
single operation, supports parameter reuse, and provides a means for tracking errors.



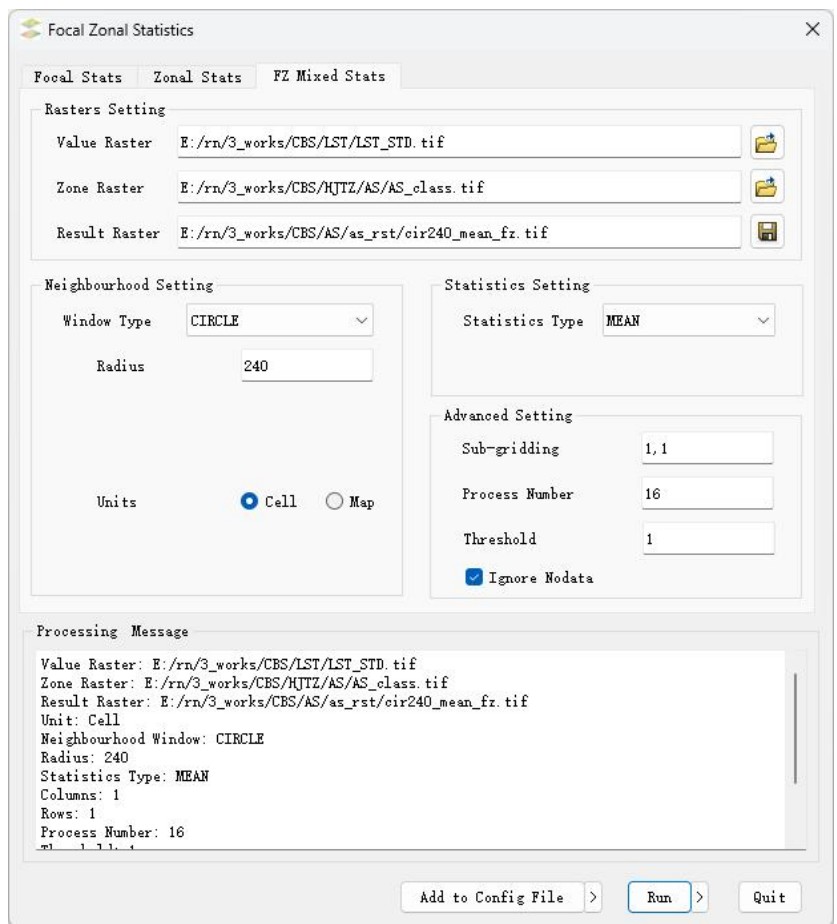


**Figure 3.** User interface design of FZStats v1.0

## 4 Experimental study

### 4.1 Background of the case

Geothermal, like coal, oil, and natural gas, is a valuable energy mineral resource, and its development and utilization play a significant role in alleviating energy supply pressure and improving the global environment (Huang and Liu, 2010; Goldstein et al., 2011). The most important indicator for geothermal resource exploration is thermal anomalies (Romaguera et al., 2018; Gemitzi et al., 2021). In recent years, with the rapid development of remote sensing, Land Surface Temperature (LST) derived from thermal infrared bands has become a key



method for identifying geothermal anomalies. However, LST is influenced by various factors,
including not only geothermal activity but also slope, aspect, and surface vegetation cover,
among other environmental factors (Tran et al., 2017; Duveiller et al., 2018; Zhao and Duan,

2020).

To effectively extract LST anomalies caused by geothermal activity, it is necessary to
suppress the influence of surface environmental variables. Within the analytical framework of
the Focal-Zonal Mixed Statistics developed in this study, terrain features are incorporated into
environmental zoning, and the spatial sliding window technique is employed to mitigate
environmental interference and enhance the abnormal information from geothermal activity.
**4.2 Data preprocessing**
**4.2.1 Spatial distribution of LST**
Landsat 8 images (Orbit Number: 116031) observed on September 16, 2013 covering the
study area, i.e., Changbai Mountain region, were used for LST mapping and geothermal
exploration in this study. After preprocessing operations such as radiometric calibration and
atmospheric correction, the Universal Single-Channel Algorithm (Jiménez-Muñoz et al., 2009,
2014; Zhang et al., 2016b) was employed to retrieve the LST of the study area, as shown in
Fig. 4. By comparing Figs. 4 and 5, it can be seen that there is a strong spatial correlation
between LST and terrain factors, especially the slope aspect. Since the local time of the
satellite passing over the study area was 10:43 AM, and the solar azimuth angle was 153°, the
LST exhibited significantly higher values on the southeast-facing slopes than on the
northwest-facing slopes.



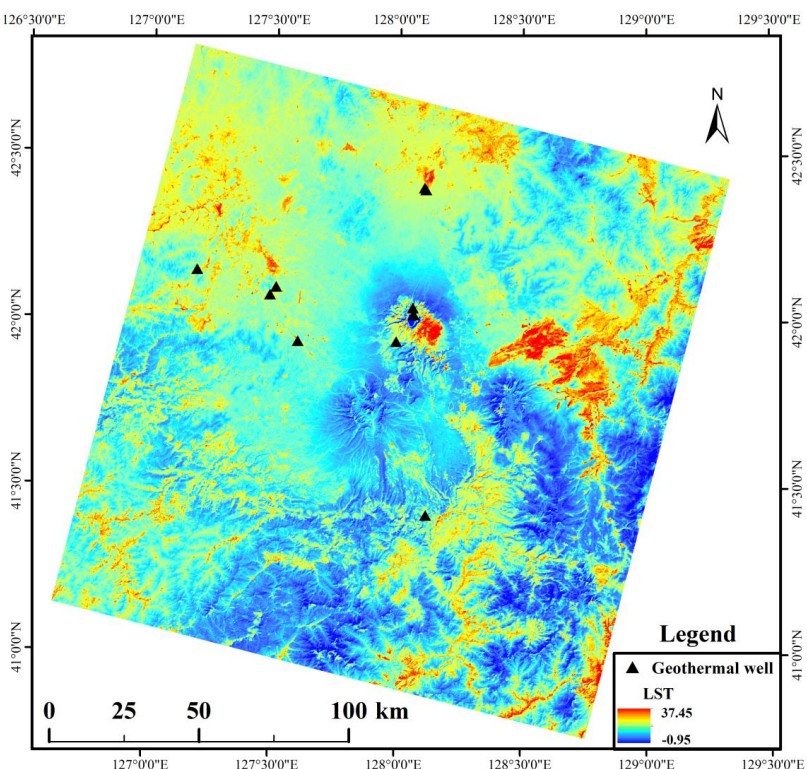

**Figure 4.** Spatial distribution of land surface temperature (LST) in the study area

**4.2.2 Mapping of unique-value environmental characteristic zones**

The slope and aspect were used as environmental factors to construct the UV-ECZR (Fig. 5a

and b). As previously mentioned, these two factors have a strong spatial coupling relationship

with LST. Although elevation and vegetation coverage were not directly applied in

environmental zoning, they can be considered similar within the neighborhood window

(Zhang et al., 2019). Therefore, their effects are indirectly suppressed. In other words, in

Focal-Zonal Mixed Statistics modeling, sample heterogeneity caused by long-range spatial

variables can be controlled through spatial proximity, while that brought by short-range

spatial variables can be suppressed through environmental similarity.





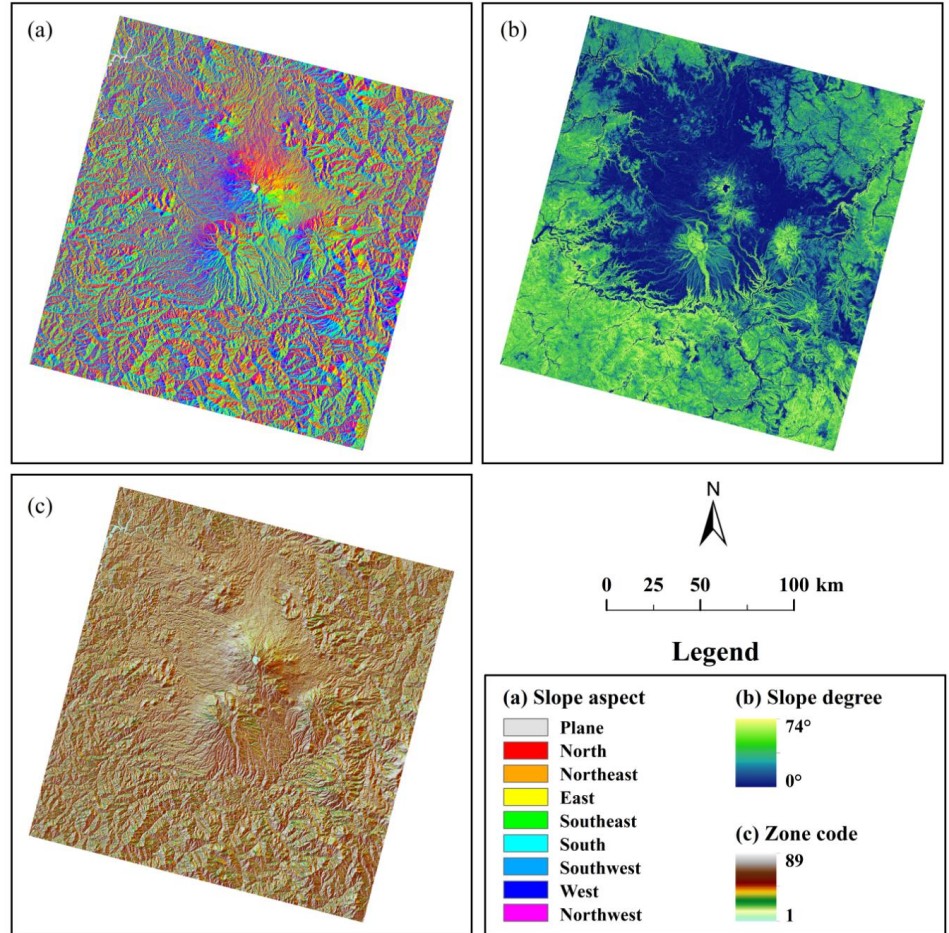

**Figure 5.** Maps of environmental factors: (a) slope aspect, (b) Slope degree, and (c) the composite

unique-value environmental characteristic zonal raster (UV-ECZR).

**4.3 Enhancement of geothermal anomalies based on Focal-Zonal Mixed Statistics**

In mineral prospectivity mapping, standard deviation standardization is often employed to

assist in constructing indicator variables for prospecting. This process involves subtracting the

mean from the original value and then dividing the result by the standard deviation. This

indicator reveals how many standard deviations the original value deviates from the mean.

The essence of this method lies in determining the appropriate range for calculating the mean

and standard deviation, enabling a comparison of the current value against the mean and using



the standard deviation to quantify this difference. In this study, Focal-Zonal Mixed Statistics
was used for this purpose, i.e., defining the comparable sample range based on both spatial
proximity and environmental similarity. Specifically, in this case, the level of LST at each
current location is assessed within the range determined by both the local window and the
similar terrain features. This approach mitigates the influence of factors such as terrain and
vegetation, thereby producing a distribution map of LST anomalies that predominantly
reflects geothermal activity.When the current window is a circle with a radius of 7.2 km, the
final enhanced geothermal anomaly is shown in Fig. 6.

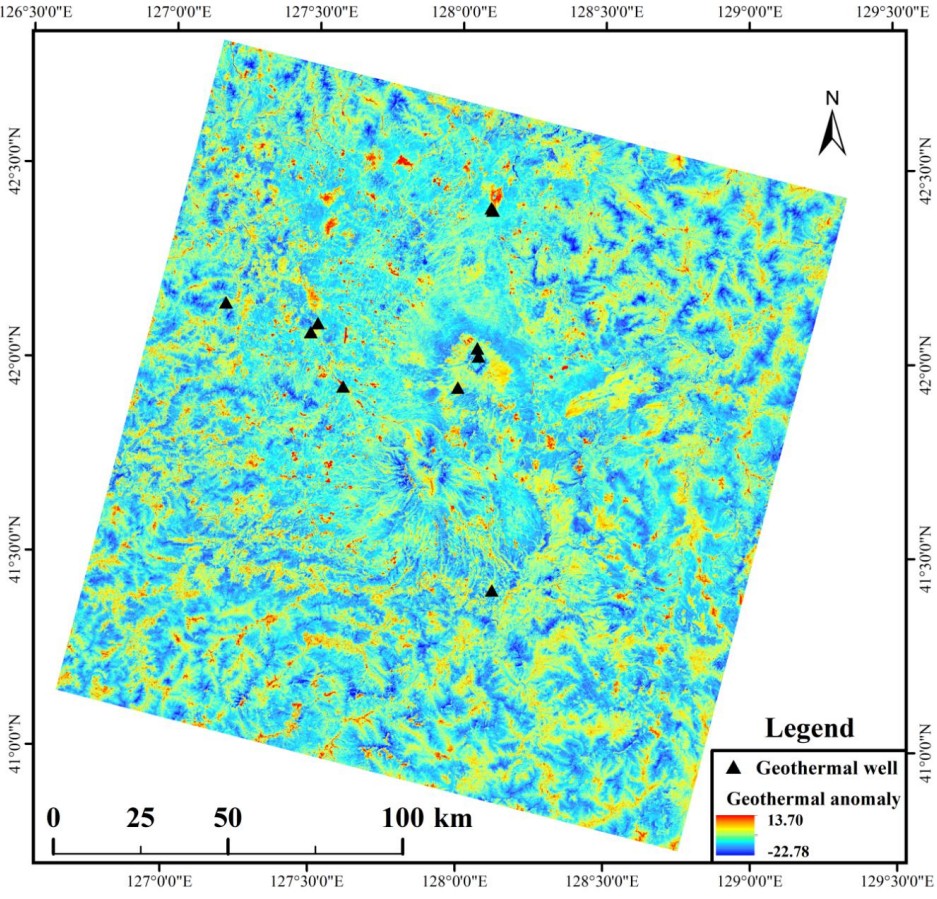


**Figure 6.** Enhanced geothermal anomaly map based on Focal-Zonal Mixed Statistics with a local window
radius of 7.2 km.

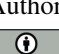



Comparing Figs. 5 and 6, it is evident that the LST anomalies enhanced using
Focal-Zonal Mixed Statistics exhibit a better spatial correlation with known geothermal wells
(obtained from Yan et al., 2017), and their high values indicate known geothermal wells more
effectively. Therefore, we have reason to believe that the high-value areas in Fig. 6 have a
higher probability of revealing new geothermal resources.
**5 Discussion**
**5.1 Advantages of the new statistics**
Based on the standard deviation standardization approach described above, we also employed
Zonal Statistics and Focal Statistics to enhance geothermal anomalies for further model
comparison. Specifically, the Receiver Operating Characteristic (ROC) curve was used to
compare the performance of LST itself and its three enhancement indices in geothermal
prospectivity mapping.
The ROC curve is plotted with the False Positive Rate (FPR) and True Positive Rate
(TPR) as the x-axis and y-axis, respectively (Fawcett, 2006; Hanczar et al., 2010), and the
resulting Area Under Curve (AUC) is used for quantitative evaluation of certain indices or
models. AUC values range from [0.5, 1], where higher values indicate better predictive
performance and accuracy of the model, and vice versa.
The ROCs of LST and its three enhancement indices obtained by Focal Statistics, Zonal
Statistics, and Focal-Zonal Mixed Statistics, respectively, are depicted in Fig. 7. It can be
observed that the enhancement effect based on Focal-Zonal Mixed Statistics is significantly
better than that based on the other two models, as the AUC of Focal-Zonal Mixed Statistics is
0.731, which is much higher than that of Zonal Statistics (0.638) and Focal Statistics (0.657).
Moreover, the AUC values of the latter two are also higher than that of LST, although
marginally.





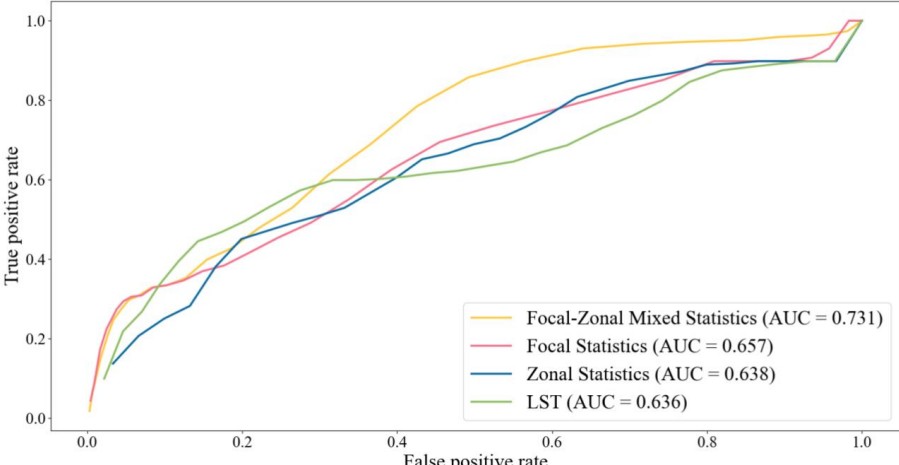


**Figure 7.** The ROCs of Land Surface Temperature (LST) and its three enhancement indicators obtained by

Focal Statistics, Zonal Statistics, and Focal-Zonal Mixed Statistics, respectively. Parameter Settings: the

local window for Focal Statistics and Focal-Zonal Mixed Statistics is a circle with a radius of 7.2 km; the

categories used for Zonal Statistics are the same as those for Focal-Zonal Mixed Statistics; and a

geothermal well represents an area of 0.1 km surrounding it.

**5.2 Robustness of the new method**

To ensure that the superior performance of the new model in Sect. 5.1 is not coincidental, it is

necessary to adjust parameters such as the local window size and the geothermal well

representative area and conduct multi-scenario comparison experiments. This will help

analyze the robustness of the new model's advantages.

When the representative area for a geothermal well is determined by 0.1km, 0.2km, and

0.3km buffers, respectively, the AUC values for LST and its enhancement indices are

calculated. These values, obtained through different models under various local window radii,

are plotted on a Cartesian coordinate system, as shown in Fig. 8.





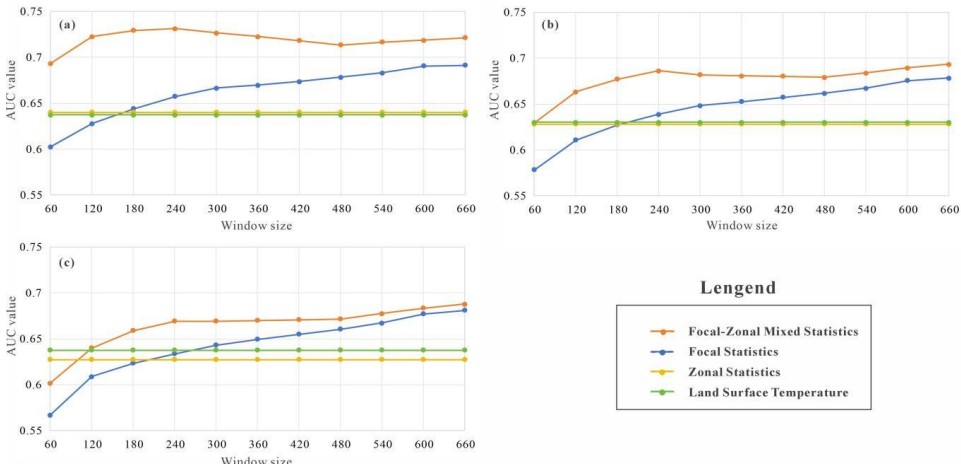

**Figure 8.** The changes in AUC values with the window size of Land Surface Temperature (LST) and its three enhancement indicators obtained by Focal Statistics, Zonal Statistics, and Focal-Zonal Mixed Statistics, when a geothermal well represents circles with a radius of (a) 0.1km, (b) 0.2km, and (c) 0.3km, respectively.

Overall, the two enhancement models incorporating neighborhood windows, i.e., Focal Statistics and Focal-Zonal Mixed Statistics, perform better than the Zonal Statistics model and the LST without enhancement. The poor performance of Zonal Statistics is due to the strong spatial variability of LST and the simplicity of the classification scheme used. Additionally, since local window methods are sensitive to spatial scale, the performance of Focal Statistics and Focal-Zonal Mixed Statistics varies with the window size. However, regardless of whether the geothermal well representative area is 0.1km, 0.2km, or 0.3km, the performance of Focal-Zonal Mixed Statistics consistently surpasses that of Focal Statistics.

**5.3 Advancements of the Toolbox**

The FZStats v1.0 developed in this study not only integrates traditional Focal Statistics and Zonal Statistics, which deal with SPD and SSH respectively, but also innovatively implements Focal-Zonal Mixed Statistics based on spatial proximity and environmental similarity, addressing both SPD and SSH. Therefore, this toolbox is expected to provide a novel solution to spatial statistics.



A variety of parameter setting interfaces are provided to enhance the statistical
applicability of the developed toolbox, ensuring it meets the requirements of different
application scenarios and computing conditions. In terms of neighborhood window settings,
in addition to rectangular and circular windows, an elliptical window is also available,
allowing users to express spatial anisotropy in the neighborhood through elliptical parameters.
Regarding statistical parameters, the new toolbox supports traditional metrics like mean,
standard deviation, minimum, and maximum values, as well as calculations for arbitrary
percentiles. To make the best use of memory and CPU capabilities, the toolbox supports raster
data chunk processing and multi-process modes, accommodating different computer memory
capacities and enabling parallel processing on multi-core CPUs. Additionally, users can set a
minimum cell number of samples for statistics through the "Threshold" parameter to avoid
low statistical precision and unreliable results due to insufficient sample size.
Lastly, to enhance automation and efficiency in multitasking, the toolbox provides a
batch processing solution. Users can write parameters into an INI-format multi-section
configuration file, which avoids repetitive and tedious manual operations. This can not only
enable one-time setup and automatic execution of multiple tasks, but support parameter reuse
and error tracing.
**6 Conclusions**
This study developed the FZStats v1.0 toolbox using Python3 and QT5. The new toolbox
integrates Focal Statistics, Zonal Statistics, and the newly developed Focal-Zonal Mixed
Statistics. We provided detailed algorithm implementations and modeling processes for these
methods and evaluated their performance in geothermal anomaly identification. The main
conclusions are as follows: First, the development of the Focal-Zonal Mixed Statistics is
essential, as it addresses gaps that traditional Focal Statistics and Zonal Statistics cannot fill.
Second, FZStats v1.0 offers extensive parameter setting options, supporting different window



shapes and types of statistics; simultaneously, by adjusting processing parameters, it can
ensure efficient performance on computers with varying configurations. Third, case analyses
show that Focal-Zonal Mixed Statistics significantly enhance geothermal anomalies compared
to Zonal Statistics and Focal Statistics methods, with this advantage being robust.
In summary, FZStats v1.0 not only innovates spatial statistical methods theoretically but
also demonstrates powerful functionality and flexibility in practical applications, making it a
promising tool in the field of geothermal anomaly identification and other areas requiring
spatial statistical solutions.
**Acknowledgments**
This study benefited from joint financial support by National Natural Science Foundation of
China (No. 42071416), the "CUG Scholar" Scientific Research Founds at China University of
Geosciences (Nos. 2022062 & 2024001).
***Code availability.*** The source code for FZStats v1.0 is available on GitHub at
https://github.com/Renna11/FocalZonalStatistics. The latest version of the software can be
obtained at https://zenodo.org/records/13208114.
***Data availability.*** The sources of the original data supporting the case study in this paper are
as follows: (1) Landsat 8 images used in this research were downloaded from
https://earthexplorer.usgs.gov (last accessed: 3 August 2024); (2) Land Surface Temperature
data can be obtained from http://databank.casearth.cn (last accessed: 3 August 2024); (3)
original elevation data used for calculating slope and aspect is from the Shuttle Radar
Topography Mission (SRTM) Global 1 Arc-Second Product, provided by NASA, available at
https://earthexplorer.usgs.gov (last accessed: 3 August 2024); and (4) geothermal well data is
sourced from Yan et al. (2017). Sample data can be found at
https://zenodo.org/records/13766015. Readers can refer to the instructions provided in the
"README.md" file on the code repository (https://github.com/Renna11/FocalZonalStatistics)



for guidance on software use, which allows for the reproduction of the case analysis using the
aforementioned original data.
***Author contributions.*** DZ and QC conceived the original idea. NR and DZ developed the
software. NR handled data processing and drafted the manuscript. DZ and QC revised the
manuscript. All authors read and approved the final manuscript.
***Competing interests.*** The authors declare no competing interests.

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
