# Peer review of "FZStats v1.0: a raster statistics toolbox for simultaneous management of spatial"

_EGUsphere, 2024_

## Author Comment (AC1)

Dear Reviewer,

We sincerely appreciate your time and valuable comments on our manuscript. Your insightful suggestions have significantly helped us improve the quality of this work. We have carefully addressed all your concerns point-by-point in the revised manuscript, which are highlighted in blue font for your convenience. The detailed responses to each comment are provided below (all suggestions and comments are colored in red).

*By using Python, this study developed a new spatial statistics toolbox named FZStats v1.0. It provides details on the development process, raw code, and a user-friendly software product. This toolbox not only includes two categories of traditional general spatial statistical tools but also integrates the new developed Focal-Zonal Mixed Statistics method, which I believe is the core contribution of this research. The manuscript is well-structured, showcasing the necessity and advantages of the proposed Focal-Zonal Mixed Statistics method through a comprehensive review of existing research, methodology, model development, and applications, with thorough discussions, making it a clearly contributive and well-written article. To further enhance the quality of this manuscript and better serve its potential readers, I offer the following suggestions for the authors' consideration:*

Thank you very much for your careful reading and encouraging comments. We are especially grateful for your constructive suggestions, which have significantly contributed to improving the quality of our work. Below, we provide point-by-point responses to each of your suggestions and comments. We sincerely hope that the revised manuscript meets your expectations and is now more informative and useful to potential readers.

*1.As instructed by the authors, the new model requires two input layers: value raster and zonal raster. In lines 214-219, the acquisition of the zonal raster is evidently based on reclassification, or in other words, the discretization of spatial variable. However,*

*how should the discretization scheme, including the number of classes and classification methods, be determined? Could the authors provide some recommendations on this? Since different reclassification parameters may significantly influence the results.*

Thank you for your insightful comment regarding the determination of the discretization scheme for the zonal raster. This is indeed a key component of the proposed methodology and deserves further clarification.

First, as a new method, the Focal–Zonal Mixed Statistics inherits the input structure and design logic of the two classical methods. From this perspective, its zonal raster is essentially consistent with the zonal raster used in traditional Zonal Statistics.

Second, from an application perspective, it is sometimes necessary to apply constraints to the construction of the zonal raster so that the resulting statistics are more accurate and meaningful.

In our specific case study of geothermal anomaly detection, we aim to identify a group of highly comparable background samples—i.e., locations that would exhibit similar surface temperature values in the absence of geothermal influence. To improve this comparability, we considered two aspects: (1) spatial proximity, which we will address in response to your next comment; and (2) environmental similarity, which is achieved through the construction of Unique-Value Environmental Characteristic Zonal Raster (UV-ECZR).

To construct the UV-ECZR, each environmental variable is first discretized, and then these layers are overlaid to form a composite zonal raster. This process involves determining both the number of classes for each variable and the appropriate classification method. Below are our recommendations:

(1) Number of Classes

A good classification scheme should aim to minimize within-zone variance and maximize between-zone variance. In practice, we found that using 5–8 classes often achieves a good balance between capturing environmental heterogeneity and maintaining sufficient sample sizes within each zone. This choice is empirical but has

proven robust in our tests.

(2) Classification Methods

For continuous variables, Natural breaks (Jenks) are suitable when the data show clear clustering. Equal interval is appropriate for uniformly distributed variables. Quantile classification ensures even representation across the value range. For categorical variables, we generally retain their original classes unless aggregation is required for specific analytical purposes.

Finally, a trade-off must be made between similarity and statistical robustness: increasing the number of classes or including too many environmental variables can yield purer environmental zones but may reduce sample sizes and model stability. Therefore, both theoretical considerations and empirical validation are necessary in setting up the zonal raster.

We have added a new description under "(2) Construction of Unique-Value Environmental Characteristic Zonal Raster (UV-ECZR)" in Section 3.1 "Modeling Process for Focal–Zonal Mixed Statistics," which provides specific recommendations on both classification methods and the number of classes. The relevant content is copied below for your reference.

*In this step, environmental factor rasters—whether continuous or categorical— are reclassified into discrete categories using a well-defined discretization scheme. For continuous variables, the classification method should be selected according to the data distribution and research objectives: natural breaks (Jenks) are recommended for datasets exhibiting clear clustering, equal interval classification suits uniformly distributed data, and quantile classification ensures balanced representation across value ranges. For categorical variables, original classes are typically retained unless aggregating categories improves analytical validity. The optimal number of classes, usually between 5 and 8, should balance environmental heterogeneity with adequate sample size within each zone. Classification performance can be evaluated by minimizing within-zone variance, maximizing between-zone variance, and assessing clustering validity through the silhouette coefficient.* (Newlines 263-273)

*2. Line 220, the configuration of model input parameters is crucial for practical applications, but the introduction to how to set and choose window size, window shape, and statistic selection is somewhat insufficient, and readers may need guidance on this aspect.*

Thank you for this important suggestion. We fully agree with your point.

Regarding the first part of your comment, as mentioned in our response to your previous question, spatial proximity is another key consideration (in addition to environmental similarity) for improving the comparability of samples in geothermal anomaly detection. The configuration of window parameters directly reflects this spatial proximity. Therefore, when setting the window size, it is essential to ensure that samples within the window are more consistent in land surface temperature (LST) compared to those outside. At the same time, a balance between similarity and statistical robustness must be considered: smaller windows tend to ensure stronger internal consistency but may result in fewer samples, reducing statistical reliability.

As for window shape, when the spatial distribution of the variable exhibits evident anisotropy, an elliptical window may be more appropriate. Otherwise, circular windows are generally recommended due to their simplicity and symmetry.

Regarding the second part of your question—the selection of the statistical measure—it should be tailored to specific application scenarios. In our geothermal case, we use the following standardized index as the statistical measure: $(T\_cell - T\_mean)$ / $SD\_window$, where the numerator reflects how much the target cell's temperature exceeds the neighborhood mean, and the denominator (standard deviation) quantifies variability within the window. This standardization improves the comparability of anomaly scores across different regions.

We have added more descriptions under "(3) Determination of neighborhood window and statistical parameters" in "Section 3.1 Modeling Process for Focal–Zonal Mixed Statistics", where we now offer concrete recommendations on window size, shape, and the selection of statistical functions for different application contexts. Please

refer to new lines 282-290 for details, as copied below.

*The window size should be selected based on several considerations, including the spatial scale of the studied phenomenon (e.g., local versus regional patterns), the resolution of the input rasters (with coarser resolution favoring larger windows), and computational efficiency (as larger windows significantly increase processing time). The window shape should be chosen according to the nature of spatial anisotropy (elliptical for directional patterns), processing efficiency (rectangular shapes are computationally faster), mitigation of edge effects (circular windows help reduce boundary artifacts), and data characteristics (rectangular for grid-aligned features and circular for isotropic phenomena).* (Newlines 282-290)

*3. Line 248, the coding method proposed by the authors is concise and clear, but there is an issue: if used over a larger area with many factor classifications, it implies more placeholders. Will using this algorithm for coding risk exceeding computer reading limits?*

We sincerely appreciate the reviewer's insightful concern regarding computational scalability. Here are our detailed explanations:

(1) Data Type Efficiency

UV-ECC values are stored as 64-bit integers (not strings), supporting up to 19 digits ($2^{64} \approx 1.8 \times 10^{19}$). For example, 8 factors with ≤10 classes each require only 8 digits—well within this limit while ensuring numerical processing efficiency.

(2) Technical Safeguards

Memory-mapped I/O: Large rasters are processed by chunks to avoid full loading.

Parallelization: Multi-core CPU support distributes computational loads.

We confirm that the method's design and implementation ensure robustness across scales. Thank you for this valuable comment!

*4. Line 392, Figure 4 lacks units.*

Thank you for your comments. We have updated Figure 4 (now it is Figure 3) by adding

the units. Please refer to the new Figure 3, which is copied below.

[Figure]

**Figure 3.** *Spatial distribution of land surface temperature (LST) in the study area on March 20, 2023.*

*5. Line 394, the authors only mention the spatial coupling of LST with slope aspect but do not empirically test whether slope and aspect are major environmental factors influencing LST. Additionally, within a 7.2 km radius, can the effect of elevation on LST be ignored, especially in relatively complex mountainous terrain?*

Thank you for your insightful comment. This is indeed an important point that we have now addressed by adding an empirical analysis of the relationship between LST and several key environmental variables, including slope, aspect, elevation, and vegetation index (NDVI). The results demonstrate that slope aspect emerges as the dominant driver

of land surface temperature (LST) variations at the local window scale, whereas elevation and NDVI exhibit less pronounced effects. This observation aligns with previous studies showing topographic orientation as a critical modulator of microclimate patterns (He et al., 2019). The underlying mechanism likely stems from two inherent properties of local landscapes:

(1) Limited terrain heterogeneity: Elevation differences within localized windows typically show constrained variability, diminishing the relative impact of absolute elevation on thermal regimes (Zhang et al., 2019).

(2) Vegetation uniformity: Vegetation composition and density tend to stabilize within small topographic units due to similar micro-environmental conditions (Yan et al., 2017).

To enhance topographic characterization, we introduced slope degree as a complementary parameter. The synergistic integration of slope aspect and slope degree achieves three critical improvements:

(1) Comprehensive terrain representation: Aspect-direction defines solar exposure patterns, while slope steepness governs surface runoff and energy retention.

(2) Microhabitat homogenization: Land units sharing equivalent aspect-slope combinations inherently exhibit reduced vegetation variability through water-energy balance constraints.

(3) Thermal regime differentiation: The aspect-degree matrix creates distinct solar radiation geometries that amplify LST contrasts between adjacent terrain units.

Accordingly, we have revised the manuscript to include these findings and have further clarified our rationale for choosing slope aspect and slope degree as the basis for constructing the Unique-Value Environmental Characteristic Zonal Raster (UV-ECZR). This decision ensures that the zonal raster reflects the most relevant terrain-driven drivers of LST variation at the target spatial scale.

*Taking the LST retrieved from the Landsat 8 image acquired on March 20, 2023, as an example, a comparison between Fig. 3 and the terrain information presented in Fig. 4 reveals a strong spatial correlation between LST patterns and topographic*

*factors, particularly slope aspect. Given that the local overpass time of Landsat 8 over the study area was approximately 11:00 AM, with a corresponding solar azimuth angle of 153°, LST values were significantly higher on southeast-facing slopes compared to northwest-facing slopes (Fig. 4a). This highlights the pronounced influence of solar radiation on the spatial variability of LST within the study area.* (Newlines 474-481)

**References**

He, J., Zhao, W., Li, A., Wen, F., and Yu, D., 2019. The impact of the terrain effect on land surface temperature variation based on Landsat-8 observations in mountainous areas. International Journal of Remote Sensing, 40(5-6), 1808-1827.

Yan, B., Qiu, S., Xiao, C., and Liang, X., 2017. Potential Geothermal Fields Remote Sensing Identification in ChangbaiMountain Basalt Area. Journal of Jilin University (Earth Sciente Edition), 47(6), 1819-1828.

Zhang, D., Xu, X., Yao, S., Zhang, J., Hou, X., and Yin, R., 2019. A novel similar habitat potential model based on sliding-window technique for vegetation restoration potential mapping. Land Degradation & Development, 31(6), 760-772.

*6. Line 423, while the mapping in the manuscript is very standardized and exquisite, some figure fonts are too small, such as in Figure 8.*

Thank you for your helpful suggestion. We have carefully revised the font sizes across all figures to ensure consistency and improved readability. Specifically, the font size in the original Figure 8 has been enlarged, and this updated figure now appears as new Figure 7 in the revised manuscript. A comparison between the old and new versions has been provided below for your reference.

[Figure]

*Old Figure 8. The changes in AUC values with the window size of Land Surface Temperature (LST) and its three enhancement indicators obtained by Focal Statistics, Zonal Statistics, and Focal-Zonal Mixed Statistics, when a geothermal well represents circles with a radius of (a) 0.1km, (b) 0.2km, and (c) 0.3km, respectively.*

[Figure]

*New Figure 7. Variations in AUC values with increasing local window radius (measured in pixel units) for Land Surface Temperature (LST) and its three enhancement indices derived from Focal Statistics, Zonal Statistics, and Focal–Zonal Mixed Statistics. The geothermal wells are represented as circles with an area of 0.035 km². Panels (a) through (d) correspond to the LST data acquired in the spring, summer, autumn, and winter of 2023, respectively.*

*7. Line 458, I am unclear about the reason for setting different representative areas for the mines.*

Thank you for your comment. The initial reason for assigning different representative areas to geothermal wells was to address potential spatial inaccuracies in their recorded locations. By using a surrounding area instead of a single pixel, we aimed to accommodate minor geolocation errors and spatial uncertainty in well positioning.

However, as shown in our previous analyses, the prediction performance does not significantly vary with changes in the representative area. This is likely because modern mapping technologies ensure reasonably accurate point locations, and clustered geothermal features are often recorded as multiple, partially overlapping points (e.g., as seen with the northernmost geothermal site in Figure. 4).

Based on these findings, in the revised version, we adopt the pixel where the geothermal well is located as the default representative area. Nevertheless, we retain both settings—a single 30m $\times$ 30m pixel (0.0009 km²) and a 0.035 km² circular area—as part of our robustness evaluation. This evaluation also includes variations across different years (2015, 2019, and 2023), seasons (spring, summer, autumn, winter), and a wide range of local window sizes (radii from 0.3 km to 9 km in 0.3 km intervals), providing a comprehensive robustness assessment.

*8. At last, after testing the software toolbox provided by the authors, I noticed that there seems to be no popup*

Thank you for your suggestion. We have added a popup notification feature to the software toolbox. Upon completion of a task, a message box will now automatically appear indicating that the process has finished, along with the total runtime. This enhancement aims to improve user experience and provide more intuitive feedback on operation status. Please see the following figure for details.

[Figure]

---

## Author Comment (AC2)

Dear Reviewer,

We are deeply grateful for your generous dedication to improving our manuscript. Your constructive and insightful comments, including those challenging questions, have been invaluable in enhancing both the quality and readability of our paper. We have carefully considered every single suggestion and implemented thorough revisions accordingly.

To facilitate your review, we have prepared a detailed PDF response document that addresses each of your comments point-by-point. In this document:

(1) Your original comments are highlighted in red;

(2) Quoted manuscript content appears in blue (with precise line numbers indicated);

(3) All modifications are clearly explained, supported by relevant equations and figures where applicable.

We sincerely hope these revisions meet with your approval. Please don't hesitate to let us know if any aspects require further clarification or improvement.

*1. This manuscript presents a Python-based tool with a graphical user interface that integrates Focal and Zonal methodologies, which are commonly used in GIS. These two approaches aggregate finer-resolution raster pixels based on different principles—Zonal methods rely on predefined rules (e.g., different topographies), while Focal methods use distance-based criteria (e.g., proximity to the center of a segment). Essentially, this tool smooths the original raster using statistical measures.*

Thank you for your comment. You are absolutely right that our method integrates both Focal and Zonal methodologies, and that the proposed model inherently includes a smoothing effect on the original raster through the use of spatial sliding windows. However, we would like to emphasize that smoothing is not the main contribution of this study.

As clarified in the revised abstract (Newlines 15‑18), the primary innovation of our Focal‑Zonal Mixed Statistics method lies in its ability to simultaneously address spatial stratified heterogeneity (typically handled by Zonal methods), positional dependence (typically handled by Focal methods), and—most importantly—the interaction between the two. This joint modeling capability fills a methodological gap in current spatial statistics tools, which generally treat these phenomena in isolation and fail to handle their co-existence in real-world spatial data.

*First, we formally develop the Focal‑Zonal Mixed Statistics model to address stratified heterogeneity, spatial dependence, and their interactions within a unified framework—filling a key methodological gap left by traditional approaches that cannot accommodate their co-occurrence in real-world spatial data.* (Newlines 15‑18)

*2. However, the manuscript lacks a clear motivation for why these specific statistics are important. Simply listing potential applications is insufficient to justify their significance. Additionally, FZStats appears to operate only on the intersection of focal and zonal zones (i.e., $( F == \text{True} \land Z == \text{True} )$), without addressing other conditions such as $( F == \text{True} \land Z == \text{False} )$ or $( F == \text{False} \land Z == \text{True} )$. This oversight raises concerns about how corner*

*cases are handled. Section 2.3 does not clarify these aspects.*

Thank you for this insightful and thought-provoking comment. It has greatly inspired us to better articulate the theoretical motivations behind our proposed method, rather than focusing solely on its practical applications.

As we briefly mentioned in our response to Comment 1, the core motivation for developing the Focal-Zonal Mixed Statistics method lies in addressing the limitations of existing Focal and Zonal approaches. Specifically:

(1) Focal methods are designed to capture *positional dependence* (Case 1: F=True, Z=False),

(2) Zonal methods handle *spatial stratified heterogeneity* (Case 2: F=False, Z=True), and

(3) The proposed method is capable of addressing *both phenomena simultaneously*, along with their interaction (Case 3: F=True, Z=True).

From a theoretical perspective, the Focal-Zonal Mixed Statistics model generalizes the two traditional methods:

(1) When the sliding window becomes infinitely large (thus rendering distance weights negligible), the method converges to Zonal Statistics, addressing stratified heterogeneity.

(2) When the zonal layer has only one unique value (i.e., no zonal partitioning), the method reduces to Focal Statistics, capturing positional dependence.

(3) Only the proposed method can accommodate the more general and realistic situation where both stratified heterogeneity and positional dependence coexist.

This capability is crucial because, in real-world spatial processes, Case 3 is often the norm rather than the exception. Neither Focal nor Zonal methods can handle this scenario alone, which underscores the theoretical necessity of our model. We have revised the abstract accordingly (Newlines 15–18), and added further elaboration in the final paragraph of the introduction section (Newlines 90–104) and the first two paragraphs in Section 5.1 (Newlines 555–567).

*Although these methods successfully integrate SSH and SPD in specific tasks such as interpolation and regression, there is still no general-purpose GIS toolbox comparable to Focal and Zonal Statistics within standard GIS workflows. To fill this*

*gap, this study presents FZStats v1.0, which unifies traditional Zonal Statistics and Focal Statistics with the novel Focal‑Zonal Mixed Statistics model. Leveraging multiprocessing and batch-processing capabilities, FZStats v1.0 improves computational efficiency and optimizes usability. Moreover, from a logical perspective, Focal‑Zonal Mixed Statistics can be viewed as a generalization of the two traditional approaches. Specifically, when the moving window covers一or substantially exceeds一 the entire study area (i.e., window size → ∞), the method converges to Zonal Statistics, effectively addressing SSH. Conversely, when only a single zone is defined, it simplifies to Focal Statistics, capturing SPD. In the more common and complex scenarios where both SSH and SPD coexist, only the mixed approach is capable of simultaneously accounting for both characteristics. We therefore anticipate that FZStats v1.0 will serve as a versatile framework for spatial analyses requiring concurrent consideration of SSH and SPD across a wide range of applications.* (Newlines 90‑104)

*Firstly, from a theoretical standpoint, traditional methods each address only one aspect of spatial variation: Focal Statistics primarily captures SPD, while Zonal Statistics is designed to account for SSH. However, real-world spatial problems often exhibit both characteristics simultaneously. This underscores the theoretical necessity and practical relevance of developing the new method—Focal–Zonal Mixed Statistics—which bridges the methodological gap between Focal Statistics and Zonal Statistics.*

*Secondly, from a conceptual perspective, Focal–Zonal Mixed Statistics can be viewed as a generalization of the two conventional approaches. When the moving window encompasses—or far exceeds—the entire study area (i.e., the window size approaches infinity), the method converges to Zonal Statistics, effectively capturing stratified heterogeneity. Conversely, when the analysis is confined to a single environmental zone, the method reduces to Focal Statistics, thereby focusing on spatial positional dependence. This flexibility enables the new method to seamlessly adapt to different spatial structures.*

(Newlines 555‑567)

Regarding your concern about whether the new model can handle F-only or Z-only scenarios, we view these as boundary or degenerate forms of our proposed model. Thus, they are not excluded, but rather naturally embedded within the broader framework. From a practical standpoint, if only one type of spatial dependence is expected (e.g., positional dependence or stratified heterogeneity alone), users may still opt for the simpler Zonal or Focal approaches—both of which are included in our FZStats v1.0 toolbox—for potentially greater computational efficiency in such specific contexts. However, this does not diminish the value of our method. The motivation behind our model is to address more generalized spatial distribution patterns, particularly when positional dependence and stratified heterogeneity coexist. In such complex scenarios where traditional methods often fall short, our approach provides a distinct advantage and broader applicability.

We have added a detailed discussion of these theoretical implications and practical use cases in the revised Section 5.1 (Newlines 555–567, as mentioned above). These additions more clearly demonstrate that our method is not just a software integration of two techniques, but rather a meaningful theoretical extension of spatial statistical modeling (Newlines 573–581 in new Section 5.1).

*Finally, regarding broader applicability, although geothermal anomaly enhancement serves as the illustrative case in this study, the utility of the proposed method extends well beyond this specific context. It is particularly well suited for applications requiring both improved sample purity and simultaneous control over SSH and SPD. Potential domains include mineral resource potential evaluation, vegetation restoration potential assessment, cropland productivity analysis, and terrestrial vegetation carbon sink estimation. Furthermore, the method can be employed to assess the spatial variability of target variables under specific environmental constraints, and to evaluate the effectiveness of environmental factors in delineating spatial patterns of interest.* (Newlines 573–581)

In summary, this work addresses a fundamental gap in current spatial statistics theory by enabling simultaneous treatment of stratified heterogeneity and positional

dependence. The development of this method is thus both theoretically justified and practically necessary.

*3. The manuscript's writing style is unconventional and contains informal expressions. For instance, "Some scholars" (Line 77) lacks citations, and "Professor Zhu and his group" (Line 79) is too informal for an academic paper. Similarly, phrases like "We believe" (Line 102) introduce unnecessary subjectivity. Figure 1 also appears redundant. Moreover, the mathematical expressions resemble descriptions of implemented Python functions rather than standard equations. It would be more appropriate to include code snippets instead. Equations 15 and 16 are not expressed in a conventional mathematical form. I recommend revising this section by representing each pixel as an indexed high-dimensional array $([CE\_1, CE\_2, ..., CE\_p])$ and grouping neighboring raster pixels based on shared label patterns.*

Thank you very much for your detailed comments and suggestions. We have thoroughly revised the manuscript's writing style, including both the specific instances you pointed out and other similar issues throughout the text. Below we respond to each item you raised:

**(1) Regarding the three examples you noted**

The expression "Some scholars have noted this issue and developed certain improved models in their respective fields to overcome the challenges posed by solely considering SSH or SPD" (Line 77) has been revised to "To address these challenges, previous studies have integrated SSH and SPD, developing specialized hybrid models for specific spatial-statistical objectives" (Newlines 78-79).

The sentence "Professor Zhu and his group expanded upon traditional spatial interpolation methods, which typically focus solely on spatial dependence, by introducing constraints derived from environmental similarity (Zhu et al., 2019)" (Line 79) has been modified to "For example, Zhu et al. (2019) extended traditional spatial interpolation methods—normally focused solely on spatial dependence—by introducing environmental similarity constraints, and formalized the "Third Law of

Geography", which states that geographically similar contexts yield similar target-variable values (Zhu et al., 2018; Zhu et al., 2020)" (Newlines 79-83)

The subjective expression "We believe that the FZStats v1.0 toolbox, especially the newly proposed Focal-Zonal Mixed Statistics, has the potential to offer methods and tools to better understand and address SSH and SPD issues" (Line 102) has been rephrased to a more neutral academic tone: "Consequently, FZStats v1.0 is positioned to function as a comprehensive analytical framework for spatial studies necessitating simultaneous evaluation of SSH and SPD parameters across diverse application domains." (Newlines 102-104).

At last, we have conducted a comprehensive review of the entire manuscript to identify and rectify similar instances of non-academic expressions throughout the paper, thereby ensuring consistency in scholarly writing standards.

**(2) Figure 1 appears redundant**

The original Figure 1 has been removed as it was identified to be redundant.

**(3) Mathematical and algorithmic expressions**

The original mathematical expressions have been modified to provide clearer distinction between theoretical descriptions and algorithmic implementations. Where more appropriate, certain formulations have been replaced with code snippets. The detailed revisions are presented below:

**Equation (1) Replacement**: The original definition of the neighborhood window in old Eq. (1) has been replaced by Listing 1. Listing 1 illustrates three types of neighborhood window classes: the rectangular window class (KDGeoRectNbhWindow), the circular window class (KDGeoCircleNbhWindow), and the elliptical window class (KDGeoEllipseNbhWindow). Additionally, the definition of the neighborhood window mask matrix has been introduced in the new Eq. (1).

```python
class KDGeoRectNbhWindow():
    def __init__(self, height: int, width: int):
        self.height = height
        self.width = width
        self.mask_matrix = self._generate_mask_matrix()

    def _generate_mask_matrix(self):...

class KDGeoCircleNbhWindow():
    def __init__(self, radius: int):
        self.radius = radius
        self.mask_matrix = self._generate_mask_matrix()

    def _generate_mask_matrix(self):...

class KDGeoEllipseNbhWindow():
    def __init__(self, semi_major_axis: int, axis_ratio: float, azimuth: float):
        self.semi_major_axis = semi_major_axis
        self.axis_ratio = axis_ratio
        self.azimuth = azimuth
        self.mask_matrix = self._generate_mask_matrix()

    def _generate_mask_matrix(self):...
```

*Listing 1. Code fragment for three types of neighborhood window classes: the rectangular window class (KDGeoRectNbhWindow), the circular window class (KDGeoCircleNbhWindow), and the elliptical window class (KDGeoEllipseNbhWindow).*

$$NM_{cx,cy}(x,y) = \begin{cases} 1 & \text{if } (x,y) \in \Omega_W \\ 0 & \text{otherwise} \end{cases} \qquad \text{New Equation} \quad (1)$$

**Equation (2) Modification**: The definition of the current focal cell's neighborhood in old Eq. (2) has been updated to explicitly define the mapping between the spatial coordinates of neighboring cells and the index positions in the neighborhood mask matrix. This revision ensures better clarity and is reflected in the new Eq. (2).

$$(x,y) = (i + u, j + v) \qquad \text{New Equation} \quad (2)$$

**Equation (3) Revision**: The old Eq. (3) defines the set of valid cells within the focal cell's neighborhood under the focal statistics framework, has been reformulated to clearly describe the spatial coordinates of neighboring cells and the constraints they must satisfy.

$$C_{F\_valid}(i,j) = \{(x,y) \in \Omega_D \mid NM_{cx,cy}(x,y) = 1\} \qquad \text{New Equation} \quad (3)$$

**Equation (4) Deletion**: The original constraint conditions for cells within a circular neighborhood window in old Eq. (4) have been removed. These constraints are now implicitly defined within the construction of the circular neighborhood window mask matrix, as shown in

Appendix Listing 1.

```python
def _calculate_mask(self):
    """
    Calculates a circular mask matrix where points within radius are marked True.
    Returns both the mask and its center coordinates.
    """
    # Determine mask dimensions
    mask_size = 2 * self.radius + 1

    # Initialize boolean matrix with False values
    mask = np.zeros((mask_size, mask_size), dtype=bool)

    # Center point coordinates of the mask
    mask_center = (self.radius, self.radius)

    # Pre-compute squared radius to avoid repeated calculation
    radius_sq = self.grid_radius ** 2

    # Iterate through each pixel in the mask
    for y in range(mask_size):
        for x in range(mask_size):
            # Check if pixel is within circular radius using distance formula
            if (y - mask_center[0]) ** 2 + (x - mask_center[1]) ** 2 <= radius_sq:
                mask[y, x] = True

    return mask, mask_center
```

*Appendix Listing 1. Construction of the circular neighborhood window mask matrix*

**Equations (5) and (6) Replacement**: The definitions of Focal Statistics and corresponding raster operations in old Eqs. (5) and (6) have been replaced by a code implementation provided in Listing 2. Listing 2 illustrates the procedure for Focal-Zonal statistical computation at a given focal cell, where a Boolean neighborhood mask is applied to filter valid values and the desired statistic is computed via a delegated statistical function.

```python
def calculate_focal_statistics_result(
    nbh_window_mask: np.ndarray,
    data_arr: np.ndarray,
    data_align_pos: Tuple[int, int],
    stats_parameters_list: List[str]
) -> float:
    # Extract neighborhood mask and data centered at the target position
    cur_nbh_mask, cur_nbh_data = calculate_current_nbh(nbh_window_mask, data_arr, data_align_pos)

    # Apply the mask to filter out invalid values
    valid_value_arr = cur_nbh_data[cur_nbh_mask]

    # Delegate the statistical computation to the external function
    return calculate_statistics(valid_value_arr, stats_parameters_list)
```

*Listing 2. Python function calculate_focal_statistics_result for computing focal statistics. The function identifies valid values from a neighborhood centered at the focal cell, filters them using a predefined mask, and then calculates the specified statistics.*

**Equations (7)-(10) Substitution with Code Implementation**: The definitions of the focal cell's zone code (old Eq. (7)), the value raster cells associated with a zone (old Eq. (8)), the statistical calculation within each zone (old Eq. (9)), and the raster-based zonal statistics operation (old Eq.

(10)) have been collectively replaced by the code implementation in Listing 3. Listing 3 illustrates the implementation of the zonal statistics procedure, where the *calculate_zonal_statistics_result* function computes statistics for each unique zone by applying the specified statistical operations on all cells within that zone.

```python
def calculate_zonal_statistics_result(
    data_arr: np.ndarray, feature_arr: np.ndarray, stats_parameters_list: List[str]
) -> np.ndarray:
    # Initialize the output array with NaNs to represent undefined statistics
    stats_result_arr = np.full_like(data_arr, np.nan)

    # Identify all unique zone codes in the feature array
    zone_code_list = np.unique(feature_arr)

    for code in zone_code_list:
        # Create a boolean mask identifying all pixels belonging to the current zone
        code_mask = (feature_arr == code)

        # Extract the data values corresponding to the current zone
        masked_data_arr = data_arr[code_mask]

        # Compute statistics for the zone
        stats_result = calculate_statistics(masked_data_arr, stats_parameters_list)

        stats_result_arr[code_mask] = stats_result

    return stats_result_arr
```

*Listing 3. Python implementation of the zonal statistics computation. The calculate_zonal_statistics_result function computes a specified statistic for each zone defined in the zone raster and assigns the result to all corresponding cells in the output raster.*

**Equation (11) Update**: The set of valid neighborhood cells under the focal-zonal framework has been redefined in new Eq. (4) to more precisely specify their spatial coordinates and the constraints they must satisfy.

$$C_{FZ\_valid}(i,j) = \{(x,y) \in \Omega_D \ | NM_{cx,cy}(x,y) = 1 \ \Lambda \ Z(x,y) = Z(i,j) \ \} \qquad \text{New Equation} \quad (4)$$

**Equations (12) and (13) Replacement**: The definitions of focal-zonal statistics and associated raster operations in old Eqs. (12) and (13) have been replaced by Listing 4, which demonstrates the implementation of the Focal-Zonal Mixed Statistics procedure. The *calculate_focal_zonal_statistics_result* integrates both spatial and zonal constraints, applies the focal-zonal mask, and computes statistics from the filtered valid values.

```python
def calculate_focal_zonal_statistics_result(
    nbh_window_mask: np.ndarray,
    data_arr: np.ndarray,
    feature_arr: np.ndarray,
    data_align_pos: Tuple[int, int],
    stats_parameters_list: List[str]
) -> float:
    # Extract neighborhood mask and data centered at the target center position
    cur_nbh_mask, cur_nbh_data = calculate_current_nbh(nbh_window_mask, data_arr, data_align_pos)

    # Extract the environmental feature values over the same neighborhood window
    _, cur_nbh_feature = calculate_current_nbh(nbh_window_mask, feature_arr, data_align_pos)

    # Retrieve the environmental feature value at the center pixel
    cur_feature = feature_arr[data_align_pos]

    # Create a mask for pixels in the neighborhood that match the center's feature value
    cur_feature_mask = (cur_nbh_feature == cur_feature)

    # Combine the neighborhood shape mask with the feature (zonal) mask
    fz_mask = cur_nbh_mask & cur_feature_mask

    # Filter data values using the combined focal-zonal mask
    valid_value_arr = cur_nbh_data[fz_mask]

    # Compute statistics only on valid data values
    return calculate_statistics(valid_value_arr, stats_parameters_list)
```

***Listing 4.*** *Python implementation of the Focal-Zonal Mixed Statistics computation. The function filters neighborhood cells based on both spatial proximity and zone code consistency, then calculates a user-specified statistic on the resulting valid subset.*

**Equation (14)**: No changes were made to the old Eq. (14), which corresponds to the new Eq. (5).

**Equation (15) Update**: The definition of UV-ECC has been modified in response to the reviewer's suggestion. The revised formulation represents each pixel as an indexed high-dimensional array and explicitly defines the computation of UV-ECC, now shown in the updated Eqs. (6) and (7).

$$\boldsymbol{CE}(i,j) = (CE_1(i,j), CE_2(i,j), \ldots, CE_p(i,j)) \qquad \text{New Equation} \quad (6)$$

$$UV - ECC \ (i,j) = \sum_{q=1}^{p} CE_q(i,j) \cdot 10^{\sum_{k=q+1}^{p} D_k} \qquad \text{New Equation} \quad (7)$$

**Equation (16) Replacement**: The original raster operation definition of UV-ECZR in the old Eq. (16) has been replaced by a code implementation provided in Listing 5. Listing 5 uses ArcPy-based raster algebra to generate UV-ECZR, where input layers represent reclassified environmental variables. A local overlay operation is performed to combine categorical codes and assign a unique zone identifier to each pixel.

```python
import os
import arcpy

feature_dir = r"E:\rn\paper\p1\A_data\f_z\L_20230928\feature"

**List of preclassified environmental variable layers (raster files)**
ce_layers = ["slope_rc9.tif", "aspect_rc9.tif"]

**Read the environmental variable layers into a list of Raster objects**
ce_rasters = [arcpy.sa.Raster(raster) for raster in ce_layers]

**Perform a cell-by-cell overlay operation (local operation in raster algebra)**
uv_eczr_raster = ce_rasters[0]
for raster in ce_rasters[1:]:
    uv_eczr_raster += raster

**Save the resulting UV-ECZR (multi-dimensional raster)**
uv_eczr_path = os.path.join(feature_dir, "slope_rc9_aspect_rc9.tif")
uv_eczr_raster.save(uv_eczr_path)
```

*Listing 5. Python implementation of UV-ECZR generation using ArcPy-based raster map algebra. Each input raster layer represents a reclassified environmental variable (e.g., slope or aspect), and the local overlay operation combines their category codes to produce a unique zone identifier for each pixel.*

**Equation (17)-(21)**: No changes were made to the old Eqs. (8) - (12).

*4. Another major concern is the lack of a systematic performance evaluation. The methodology is benchmarked against a single real-world dataset with cross-sectional environmental variables, disregarding the fact that such variables often form time series. To strengthen the evaluation, the authors should incorporate multiple benchmarking datasets. Aside from the AUC metrics, Figure 7 suggests that LST outperforms FZStats within the FPR range of approximately 0.1–0.3, raising questions about the legitimacy of the FZStats approach, because methods with FPR beyond this range may not be desired. The manuscript entirely neglects the trade-off between TPR and FPR, which is critical for assessing the method's robustness.*

Thank you for your comment. We have further enhanced the robustness evaluation. In addition to the original considerations of local window size and representative area of geothermal wells, we have now incorporated multi-temporal image analysis.

In the revised manuscript, Landsat 8 imagery (Orbit Number: 116031) from the

spring, summer, and autumn seasons of 2015, 2019, and 2023 was used for land surface temperature (LST) retrieval and geothermal anomaly extraction in the Changbai Mountain region. This multi-temporal and multi-seasonal data selection not only enables a more comprehensive assessment of the method's effectiveness and robustness, but also facilitates the identification of the optimal season for geothermal anomaly detection, thereby supporting more informed geothermal exploration.

The following description has been added in "Section 4.2.1 Spatial distribution of LST":

*In this study, Landsat 8 imagery (Orbit Number: 116031) acquired during the spring, summer, and autumn seasons of 2015, 2019, and 2023, covering the Changbai Mountain region, was utilized for land surface temperature (LST) mapping and geothermal anomaly detection. The selection of multi-temporal images across different seasons and years was intended to robustly validate the effectiveness of the proposed method and to explore the temporal evolution patterns of geothermal anomalies, thereby providing improved support for geothermal exploration.* (Newlines 464-469)

In addition, by adjusting the size of the local window and the representative area of geothermal wells, the new method demonstrated improved performance in ROC curve analysis, especially in terms of its accuracy in predicting extreme thermal anomalies (as you mentioned that is "within the FPR range of approximately 0.1–0.3") compared to traditional methods. Please refer to the revised Figure 6.

[Figure]

**Figure 6.** *Receiver Operating Characteristic (ROC) curves of the Land Surface Temperature (LST) and its three enhancement indicators derived from Focal Statistics, Zonal Statistics, and Focal‑Zonal Mixed Statistics, respectively. A Parameter settings: the local window used for both Focal Statistics and Focal‑Zonal Mixed Statistics is a circle with a radius of 4.2 km; the zoning categories used for Zonal Statistics are identical to those employed in Focal‑Zonal Mixed Statistics; and a geothermal well represents an area of 0.035 km2 surrounding it.*

Furthermore, several additional observations can be drawn from Figure 6:

*...comparison of Fig. 6a‑d indicates that our enhanced model performs best in autumn, as evidenced by the highest AUC value observed in this season.*(Newlines 544-545)

*5. Overall, I am not convinced that this paper meets the publication standards of GMD in its current form. Substantial revisions are necessary to improve the motivation, methodology, and evaluation:*

Thank you for your comment. Your observations regarding the motivation, methodology, and evaluation are highly constructive.

We have conducted comprehensive revisions accordingly in three key parts of the manuscript: the new Abstract and "1 Introduction", the revised "2 Models" and "3 Module Design", and the updated "4 Experimental Study" and "5 Discussion". These revisions aim to better clarify the theoretical rationale, improve methodological rigor, and enhance the comprehensiveness of the empirical evaluation.

First, regarding the motivation, we have clarified the theoretical foundation of this study. Specifically, we frame spatial positional dependence and stratified heterogeneity within a unified perspective. Although traditionally treated separately—corresponding to focal and zonal statistical methods respectively—these two forms of spatial variation often coexist in real-world applications. We argue that the proposed method generalizes both classical approaches, which can be seen as special cases of our model. This highlights the necessity and theoretical value of developing a unified method that accommodates a broader range of spatial analysis needs.

Second, with respect to the methodology, we have thoroughly revised the text, mathematical formulations, and figures to improve clarity, scientific rigor, and readability. The expressions now better distinguish theoretical constructs from algorithmic implementations, making the paper more accessible to both academic and practical audiences.

Finally, in terms of evaluation, we expanded our experiments to include multi-temporal Landsat imagery from different years and seasons. This allowed for a more robust land surface temperature (LST) inversion and geothermal anomaly detection. These selections of years, seasons, neighborhood sizes, and point representativeness were all deliberately designed to evaluate the stability and generalizability of the proposed method relative to the two traditional approaches. This ensures that the conclusions drawn are well-grounded and applicable across diverse conditions.